# The 8:1:1 Supplementation of Branched-Chain Amino Acids in High-Intensity Training: A Case Study of the Protective Effect on Rhabdomyolysis

**DOI:** 10.3390/healthcare12080866

**Published:** 2024-04-22

**Authors:** Angel Vicario-Merino, Marcos A. Soriano, Ester Jiménez-Ormeño, Carlos Ruiz-Moreno, Cesar Gallo-Salazar, Francisco Areces-Corcuera

**Affiliations:** 1Mountain Care and Inhospitable Environments Research Group, Department of Nursing, HM Hospitals Faculty of Health Sciences of the UCJC, University Camilo José Cela, C/Castillo de Alarcón, 49, 28692 Madrid, Spain; 2Strength Training and Neuromuscular Performance Research Group (StrengthP_RG), Department of Physical Activity and Sports Sciences, HM Hospitals Faculty of Health Sciences of the UCJC, University Camilo José Cela, C/Castillo de Alarcón, 49, 28692 Madrid, Spain; masoriano1991@gmail.com (M.A.S.); ejimenez@ucjc.edu (E.J.-O.); cgallo@ucjc.edu (C.G.-S.); fareces@ucjc.edu (F.A.-C.); 3Exercise Physiology Laboratory, Department of Physical Activity and Sports Sciences, HM Hospitals Faculty of Health Sciences of the UCJC, University Camilo José Cela, C/Castillo de Alarcón, 49, 28692 Madrid, Spain; cruizm@ucjc.edu

**Keywords:** BCAA, rhabdomyolysis, protection, muscle pain, RPE

## Abstract

Introduction: The increasing prevalence of high-intensity sports activities, notably the burgeoning popularity of CrossFit, underscores the contemporary significance of such physical pursuits. The discernible protective impact of branched-chain amino acids on muscle fatigue and injuries is emerging as a noteworthy area of investigation. Within the realm of sports, integrating BCAA supplementation into dietary practices holds promise for aiding athletes in their recovery, particularly in mitigating Delayed-Onset Muscle Soreness. Methodology: This study adopted an experimental pilot design with repeated measures, employing a controlled and randomized approach through double-blind procedures. The participant engaged in high-intensity activity, specifically the CrossFit Karen^®^ test, which entailed executing 150 wall ball throws (9 kg) to a height of 3 m. The trial incorporated three randomized supplementation conditions: BCAAs in an 8:1:1 ratio or a 2:1:1 ratio or a placebo condition. The participant consumed 15 g daily for 7 days, commencing 72 h prior to the initial blood sample and the first Karen^®^ test. Results: In this study, BCAA supplementation at an 8:1:1 ratio demonstrated a discernible protective effect against muscular damage, as evidenced by creatine kinase values and ratings of perceived exertion.

## 1. Introduction

High-intensity sports activities are gaining prominence in society, with CrossFit being one of the most rapidly growing activities in recent years [1]. It is a high-intensity activity which, due to the increasing number of participants and the lack of control and monitoring, is generating various types of injuries among practitioners [2]. 

In addition to the potential injuries they cause, high-intensity activities have a direct impact on physiological and hematological aspects, resulting in elevations of enzymes such as creatine kinase (CK). Traditionally, this enzyme has been used to assess muscle damage caused by intense physical activity and as an indicator of elements related to pain and metabolic injury among athletes [3].

The risk of rhabdomyolysis appears when the values of CK in blood increase over 1000 uL^−1^ or when the values are five times greater than the baseline, which are representative values of muscular damage with an onset between 24 and 48 h post exercise. The values of myoglobin could also be used, as their peak value appears more rapidly, but it is metabolized quicker, making the values of CK more reliable [4].

Recently, the protective effect of branched-chain amino acids (BCAAs) on muscle fatigue and injuries has been becoming evident [5]. In a sports context, including BCAA supplementation in the diet could help athletes in recovery, at least in reducing Delayed-Onset Muscle Soreness (DOMS). Thus, this may be an effective strategy for coaches to manage workload and muscle damage. However, the scientific evidence regarding muscle damage recovery, specifically in relation to creatine kinase (CK) levels, is inconsistent. This inconsistency may be due to the existing discrepancy in BCAA dosage, which is typically used in a 2:1:1 ratio. Therefore, according to the authors’ knowledge, the effectiveness of higher doses of BCAAs has not yet been experimentally explored. The objective of this pilot study was to examine variations in CK levels in an athlete after exposure to high-intensity activity (CrossFit Karen^®^ protocol) and, through a double-blind study, determine whether supplementation with branched-chain amino acids has any impact on key indicators of pain, fatigue, and recovery among athletes.

## 2. Materials and Methods

### 2.1. Inclusion Criterion

In this experimental pilot study with repeated measures, controlled and randomized by double-blind procedures, a 34-year-old male voluntarily participated; he weighed 74 kg and had a standing at a height of 169 m, with a body mass index of 25.9 (above-normal muscle mass). The VO2max of the subject was calculated using the Bruce test, being an indirect method with higher validity, obtaining a value of 61.5 mL/kg/min [6]. 

Inclusion criteria for the pilot study were more than 6 months of experience in high-intensity training, with a minimum frequency of 3 times per week. Exclusion criteria were previous injuries in the last 3 months, chronic diseases or renal problems, and being under any treatment during the research period.

### 2.2. High-Intensity Activity

The participant underwent high-intensity activity, specifically the Crossfit Karen^®^ test [7], consisting of 150 wall ball throws (9 kg) to a height of 3 m. The standardized Crossfit Karen^®^ test warm-up was carried out previously. During the test, a researcher verified the correct technical execution, recording poorly executed attempts and counting the correct ones until the goal of 150 was reached.

This test was conducted under three randomized supplementation conditions, with 11 days between the first two sample collections and 26 days between the second and third data collection. The protocol remained consistent across all conditions: the initiation of supplementation one week before the first session involving blood sample extraction and the performance of the test.

### 2.3. Supplementation

The supplementation was provided by Life Pro Nutrition industries, (Madrid, Spain) and prepared by Indiex Laboratories, offering three formulations: BCAA in an 8:1:1 ratio or a 2:1:1 ratio (with l-leucine being the highest in proportion) or a placebo preparation where the amino acids were substituted with maltodextrin and rice flour. The participant took 15 g per day, starting 72 h prior to the first blood sample and the first Karen^®^, continuing the intake throughout the week, adding up a total of 7 intakes, with all of them at the same time of day. 

Following the recovery time study protocol for trained athletes performing the CrossFit Karen routine [7], the participant commenced supplementation 7 days before the initial data collection. Blood samples were collected before the Karen^®^ test at 24, 48, and 72 h, with CK values in blood analyzed. The rate of perceived exertion (RPE Scale 0–10) [8] was recorded at each sampling point using the Visual Analog rating scale [9].

To prevent potential biases, the participant was instructed to adhere to the following instructions:Suspend their training routines.Rest for 24 h prior to the Karen^®^ test.After the first sample collection, refrain from practicing the same exercise (to avoid memory effects and increased resistance).Not consume any medication (including pain relievers), alcohol, or food products containing stimulating substances such as caffeine (coffee, tea, cola drinks, etc.).Maintain a consistent type of diet during the week in which the research was carried out. The diet consisted of controlling protein intake so that it did not exceed the amount of 2 g/kg and trying to eat similar foods during the week in which the participant took part in the research.

### 2.4. Obtention of Samples

Blood samples were obtained at the sports center, centrifuged post-extraction, and stored in a temperature-controlled refrigerator, always maintaining the cold chain until transportation to the clinical analysis center for testing.

Before participation, the individual was informed about the study and provided informed consent. The research protocol adhered to the principles of the Declaration of Helsinki of the World Medical Association and received approval from the ethics committee of the Universidad Camilo José Cela (10_22_EPR_BCAA, 21 December 2022).

## 3. Results

The CK and RPE values of the participant for each experimental condition in this study are shown in Table 1.

The results obtained show how the RPE and the CK values varied depending on the samples taken: CK values decreased from sample 0 and onwards, as did the RPE values, with the 8:1:1 preparation (Figure 1). 

In this subject, CAA supplementation with the ratio of 8:1:1 proved to have a protective effect on muscular damage, as seen in the CK values and in the RPE, allowing the subject to reach a perception of no pain 72 h later. 

## 4. Discussion

In this case study, CK values showed a high peak in the 24 h samples in the placebo and 2:1:1 conditions, which decreased substantially in the 48 h and 72 h samples. However, in the BCAA 8:1:1 supplementation condition, these values were maintained in a similar way in the first two samples (0 and 24 h), decreasing progressively and without showing high values. Furthermore, RPE values showed high values in three first samples in the placebo and 2:1:1 conditions, while in the 8:1:1 condition, these values were considerably lower in the 24, 48 and 72 h samples.

The results depicting an increase in CK values demonstrate how the muscular damage suffered by the subject was observed to clearly change in the values at 24 h. This variation was particularly noteworthy in the placebo and 2:1:1 condition, aligning with marked CK value fluctuations reported in similar studies involving trained athletes [10].

In CrossFit exercise protocols (Work Of the Day, WOD), it seems that repetitive eccentric overloading increases muscular damage, highlighting an elevation in the biochemical marker of CKs. Interestingly, in a study where they assessed CK markers at baseline, 24, 42, and 72 h after a CrossFit Karen^®^ protocol, the CK levels 24 h later increased by 54.55% [7]. In this case study, the 0–24 h CK values varied between −6.8% for the 8:1:1 BCAA sample and 583.8% and 668.77% for the placebo and 2:1:1 samples, respectively. Therefore, the effectiveness of the Karen protocol in generating muscular damage and the effects of the BCAA 8:1:1 preparation appear evident.

Substantially elevated CK values expose athletes to the risk of rhabdomyolysis [11], which is already a sufficiently serious and currently asymptomatic pathology to be considered. The potential emergency medical risk lies in the required treatment for an athlete in case of an accident, as the established administration of NSAIDs [12,13] for pain management may prove insufficient, potentially leading to increased renal impairment. The risk escalates in athletes self-medicating for pain arising from activity through the indiscriminate use of NSAIDs [14], consequently increasing the risk of renal failure [15].

In vivo/ex vivo experiments with BCAAs, researchers have clearly identified the effects on muscle atrophy and in the myofiber cross-sectional area as well as muscle force and its compliance to stress and are looking into the study of optimized BCAAs to improve its biodistribution and its effects on mammals with a 2:1:1:2 ratio (BCAAS + 2 L-Alanine). Although the protocol with a 2:1:1 ratio did not seem protective, this may be because studies demonstrating its effectiveness typically involve a supplementation duration of around 10 days. We supplemented the participant for 7 days, with the 5th day showing the highest increase in CK levels. Therefore, an 8:1:1 ratio appears to be a prophylactic dose when the supplementation time is less than 10 days, acting in compensation with lower doses [16].

The data obtained reinforce the findings of studies associating repetitive eccentric activities with excessive elongation, which distorts the sensory area of nerve endings [17]. Such repetitively and intensively performed exercises can potentially entrap nerve endings in the muscle spindle, contributing to microlesions in nerve endings generated by the activity undertaken [18]. Different and early-stage lines of investigation propose leucin (one of the amino acids in the BCAA mix) to provide a protective effect on the muscle [19] and IKVAV composed of isoleucine, lysine, and valine as functional motifs with promising preliminary results in providing a positive effect on potential nerve regeneration [20]. All of these effects may explain the results obtained in this subject [21].

The obtained values support studies on the protective effect of BCAA supplementation with an 8:1:1 ratio in reducing muscular damage [7]. Exercise with placebo supplementation or with low BCAA supplementation values, such as 2:1:1, does not exhibit a protective effect on CK values. 

### Limitations to the Case Study

The presented case study took an ecological approach, maintaining the nutritional routines of the subject. The only requested modification was to include nutritional supplementation with BCAAs following the study protocol. There was no in-depth control or analysis of the individual nutritional parameters; therefore, the results of this case study must be interpreted with caution. 

## 5. Conclusions

With the results of this case study, it can be proposed that the 8:1:1 ratio of BCAA supplementation may provide a protective effect on pain and on CK values, as it acted as a protective supplement for the subject in this study. 

## Figures and Tables

**Figure 1 healthcare-12-00866-f001:**
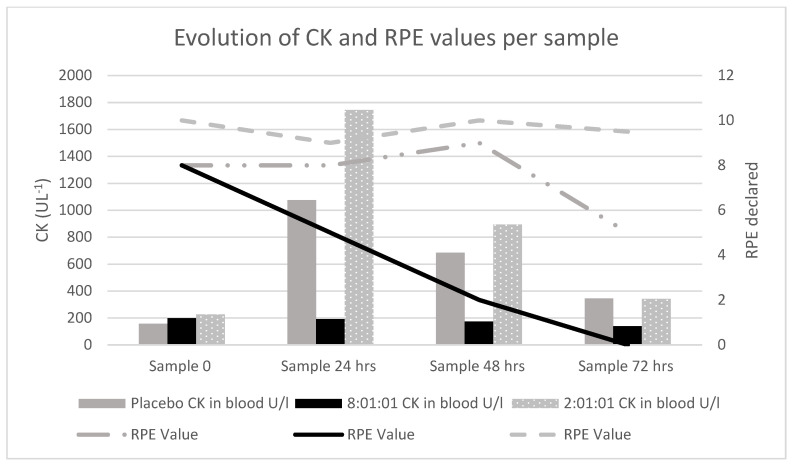
Evolution of the creatine kinase (CK) and rate of perceived exertion (RPE) values depending on the sample.

**Table 1 healthcare-12-00866-t001:** Creatine kinase (CK) and rate of perceived exertion (RPE) values obtained for each experimental condition.

	Placebo		8:1:1		2:1:1	
	CK in Blood (U/L)	RPE	CK in Blood (U/L)	RPE	CK in Blood (U/L)	RPE
Sample 0 (post test)	157.4	8	199.4	8	227.0	10
Sample 24 h	**1076.3**	8	192.6	5	**1745.1**	9
Sample 48 h	684.4	9	173.4	2	894.3	10
Sample 72 h	344.9	5	139.0	0	342.1	9.5

## Data Availability

The raw data supporting the conclusion of this article will be made available by the authors without undue reservation.

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
