# Peer review of "The 8:1:1 Supplementation of Branched-Chain Amino Acids in High-Intensity Training: A Case Study of the Protective Effect on Rhabdomyolysis"

_healthcare, 2024, doi:10.3390/healthcare12080866_

Round 1
Reviewer 1 Report
Comments and Suggestions for Authors
The present study evaluated the effect of high doses of BCAAs in a placebo-controlled case study in pilot research with CrossFit practitioners. The idea of the study is very interesting, well-aimed at potential innovations that new sports modalities can bring opportunities concerning Sports Nutrition. The authors took care to send the biochemical exam reports for each day as supplementary materials, which makes the quality of the investigation even more robust. Despite this, I leave below some questions and suggestions with the intention of improving the quality of the case study:
1. Separate materials and methods into subtopics. It is very disorganized, and the authors have a wealth of data that could be used for future studies with greater statistical power in the future if the methodology used is well presented.
2. Inform the brand of BCAA used and the factory.
3. Informing which amino acid is in each proportion, it could be the more concentrated L-leucine, L-isoleucine or L-valine, which can confuse the reader.
4. Researchers require the inclusion of dietary data (current consumption) of key macro- and micronutrients (iron, vitamin D, calcium, at least) for the period before, during and after the CrossFit Karen test. This is the most vulnerable point of the work, as it can directly affect the findings as a confounding factor.
5. Along the same lines, I strongly recommend including the RED-S scale (Relative Energy Deficiency Scale) quantified for these days in kcal/kg of lean mass/day as a parameter of energy availability. It is associated with a value < 30.0 kcal/kg/day with the incidence of muscle damage and rhabdomyolysis, that is, it would be an important addition to the quality of the case study.
6. In Figure 1 the authors use RPE and EPR, please use the same acronym for the same parameter.
7. Do the authors have a pain assessment (visual analogue scale) for these cases? It would be an interesting addition to the case study.
8. What was the placebo condition made up of? This does not appear in the manuscript.
9. Was the participant using medication? Food supplements? This is not described in the manuscript.
10. Do the authors have an estimate (even indirect) of the participant's VO2max? I strongly recommend including the information to better characterize the case.
Author Response
We appreciate reviewer’s suggestions. Their comments and suggestions have greatly impacted the quality of this manuscript.
Please find the answers and comments below:
- Separate materials and methods into subtopics. It is very disorganized, and the authors have a wealth of data that could be used for future studies with greater statistical power in the future if the methodology used is well presented.
Done. Please see update in different color (purple)
- Inform the brand of BCAA used and the factory.
The brand of BCAA is Lifepro Nutrition. Updated in the manuscript.
- Informing which amino acid is in each proportion, it could be the more concentrated L-leucine, L-isoleucine or L-valine, which can confuse the reader.
The proportions were: L-leucine (2 or 8), L-isoleucine (1) or L-valine (1). Updated in the manuscript.
This information is available in the informed consent that the participant had to sign. See the extract where this is explained: “The present study aims to assess the effectiveness of supplementation with 12 g/d for 7 days of branched chain amino acids (BCAA) in a proportion of (1:1:2) (valine: isoleucine: leucine) or (1:1 :8) (valine: isoleucine: leucine) from the LIFEPRO nutrition brand versus a placebo to improve performance and prevent muscle damage in a Karen crossfit protocol.”
- Researchers require the inclusion of dietary data (current consumption) of key macro- and micronutrients (iron, vitamin D, calcium, at least) for the period before, during and after the CrossFit Karen test. This is the most vulnerable point of the work, as it can directly affect the findings as a confounding factor.
This information is available in the informed consent that the participant had to sign. See the extract where this is explained: “It is essential that during the study you do not consume any medication (including pain relievers), alcohol, or food products containing stimulating substances such as caffeine (coffee, tea, cola drinks, etc.). Furthermore, you must maintain a diet type the week of the research. The diet consists of controlling protein intake that does not exceed the amount of 2 g/kg and trying to eat similar foods during the weeks in which you participate in the research.”
Updated a remark in the manuscript.
- Along the same lines, I strongly recommend including the RED-S scale (Relative Energy Deficiency Scale) quantified for these days in kcal/kg of lean mass/day as a parameter of energy availability. It is associated with a value < 30.0 kcal/kg/day with the incidence of muscle damage and rhabdomyolysis, that is, it would be an important addition to the quality of the case study.
This has been answered in the previous answer.
- In Figure 1 the authors use RPE and EPR, please use the same acronym for the same parameter.
Thank you for your advice. In accordance with your instructions, this has been corrected.
- Do the authors have a pain assessment (visual analogue scale) for these cases? It would be an interesting addition to the case study.
Manuscript updated with the information requested. The scale used was the Visual Analog and Graphic Rating Scales for Assessing Pain Following Delayed Onset Muscle Soreness.
Reference:
Mattacola, C., Perrin, D., Gansneder, B., Allen, J., & Mickey, C. (1997). A Comparison of Visual Analog and Graphic Rating Scales for Assessing Pain Following Delayed Onset Muscle Soreness. Journal of Sport Rehabilitation, 6, 38-46. https://doi.org/10.1123/JSR.6.1.38.
- What was the placebo condition made up of? This does not appear in the manuscript.
The placebo supplementation had the exact same ingredients as the other preparations, but the amino acids were substituted with Maltodextrin and rice flour.
Updated a remark in the manuscript.
- Was the participant using medication? Food supplements? This is not described in the manuscript.
The consent form document that the participant had to sign prior to the inclusion in the study included this information. The translated text taken from the consent form is:
It is essential that during the study you do not consume any medication (including pain relievers), alcohol, or food products containing stimulating substances such as caffeine (coffee, tea, cola drinks, etc.). Furthermore, you must maintain a diet type the week of the research. The diet consists of controlling protein intake that does not exceed the amount of 2 g/kg and trying to eat similar foods during the 2 weeks in which you participate in the research.
An annotation clarifying this has been included.
- Do the authors have an estimate (even indirect) of the participant's VO2max? I strongly recommend including the information to better characterize the case.
The indirect VO2 max of the subject is: 61.5ml/kg/min.
Calculated using the Bruce indirect method:
- Peric and Z. Nikolovski, “Validation of four indirect VO2max laboratory prediction tests in the case of soccer players,” Journal of Physical Education and Sport, vol. 17, no. 2, p. 608, 2017, doi: 10.7752/jpes.2017.02092.
Reviewer 2 Report
Comments and Suggestions for Authors
A very interesting study suggesting the positive effect of BCAA supplementation (especially 8:1:1) on alleviating the effects of high-intensity training.
The following comments should be taken into account:
1. Make the introduction more specific by shortening or eliminating the argument about injuries (up to line 49).
2. Consider whether the classification of the athlete to the standards of non-training people in the "overweight (25.9)" category is false, as it probably results from "above normal" muscle mass.
3. Consider whether we should talk about improving muscle regeneration rather than rhabdomyolysis.
Author Response
We appreciate reviewer’s suggestions. Their comments and suggestions have greatly impacted the quality of this manuscript.
Please find the answers tot he comments below:
- Make the introduction more specific by shortening or eliminating the argument about injuries (up to line 49).
Thank you for the suggestion. Updated.
- Consider whether the classification of the athlete to the standards of non-training people in the "overweight (25.9)" category is false, as it probably results from "above normal" muscle mass.
Thank you for the suggestion. Updated.
- Consider whether we should talk about improving muscle regeneration rather than rhabdomyolysis.
Thank you very much for such appreciation. The reason why we have focused it on rhabdomyolysis is because the high values of CK were surprisingly high and the health of the high-intensity sportsmen and women became a concern.
Having said this, when performing the reference research, that there can be a side effect of muscle protectiveness and / or regeneration, which is remarked in the discussion, but we did and do not have enough evidence to be able to confirm this.
Round 2
Reviewer 1 Report
Comments and Suggestions for Authors
The authors answered almost all of my comments in the first round. I would like to thank all for this revised version. However, after reading all of the answers I believe that not all of the points are adequately addressed.
For this reason, I would like to suggest a possible addition: the current study is interesting, but lacks the major dietetic control endpoints for a robust study using muscle soreness. Only using the description "It is essential that during the study you do not consume any medication (including pain relievers), alcohol, or food products containing stimulating substances such as caffeine (coffee, tea, cola drinks, etc.). Furthermore, you must maintain a diet type the week of the research. The diet consists of controlling protein intake that does not exceed the amount of 2 g/kg and trying to eat similar foods during the 2 weeks in which you participate in the research." does not guarantee that the participant effectively maintained a normocaloric intake during the study. So, this could be the reason for interesting findings, not the BCAA supplementation.
Despite this main problem, I would suggest that the authors include a section at the final of the discussion, as well as in the conclusions of the study, that due to not analyzing nutritional parameters, the investigation must be interpreted with caution and could have been a synergistic effect of BCAA supplementation with the dietary pattern during the study period. The reader must be aware of this scenario to interpret adequately the findings of this study.
Author Response
Dear reviewer,
Following your recommendations we have added the section 4.1 Limitations to the case study and updated the last part of the conclusions to specify that these results may have a protective effect on the subject in the study.
Please see the added section and modifications in a different color.
Thank you for your recommendations to enhance and improve the manuscript.
Best regards.